# Childhood Brain Tumors: A Review of Strategies to Translate CNS Drug Delivery to Clinical Trials

**DOI:** 10.3390/cancers15030857

**Published:** 2023-01-30

**Authors:** Ruman Rahman, Miroslaw Janowski, Clare L. Killick-Cole, William G. B. Singleton, Emma Campbell, Piotr Walczak, Soumen Khatua, Lukas Faltings, Marc Symons, Julia R. Schneider, Kevin Kwan, John A. Boockvar, Steven S. Gill, J. Miguel Oliveira, Kevin Beccaria, Alexandre Carpentier, Michael Canney, Monica Pearl, Gareth J. Veal, Lisethe Meijer, David A. Walker

**Affiliations:** 1Children’s Brain Tumor Research Centre, Biodiscovery Institute, University of Nottingham, Nottingham NG7 2RD, UK; 2Center for Advanced Imaging Research, Department of Diagnostic Radiology and Nuclear Medicine, University of Maryland, 655 W. Baltimore Street, Baltimore, MD 21201, USA; 3Functional Neurosurgery Research Group, Translational Health Sciences, Bristol Medical School, University of Bristol, Level 1 Learning and Research Building, Southmead Hospital, Westbury-on-Trym, Bristol BS10 5NB, UK; 4Clinical Neurosciences, Translational Health Sciences, University of Bristol & Bristol Royal Hospital for Children, Level 1 Learning and Research Building, Southmead Hospital, Westbury-on-Trym, Bristol BS10 5NB, UK; 5Department of Pediatrics Hematology/Oncology, Mayo Clinic/Rochester Cancer Center, 200 First St. SW, Rochester, MN 55905, USA; 6Brain Tumor Biotech Center, Department of Neurosurgery, Lenox Hill Hospital, Zucker School of Medicine at Hofstra/Northwell, 100 E 77th Street, New York, NY 10075, USA; 7Feinstein Institute for Medical Research, Zucker School of Medicine at Hofstra/Northwell, 350 Community Dr, Manhasset, NY 11030, USA; 83B’s Research Group, I3Bs–Research Institute on Biomaterials, Biodegradables and Biomimetics, University of Minho, Headquarters of the European Institute of Excellence on Tissue Engineering and Regenerative Medicine, AvePark, Parque de Ciência e Tecnologia, Zona Industrial da Gandra, Barco, 4805-017 Guimarães, Portugal; 9ICVS/3B’s-PT Government Associate Laboratory, Largo do Paço, 4704-553 Braga, Portugal; 10APHP, Department of Pediatric Neurosurgery, Necker-Enfants Malades Hospital, 149 Rue de Sèvres, 75015 Paris, France; 11Carthera, Laënnec Bioparc, 60A Avenue Rockefeller, 69008 Lyon, France; 12Department of Diagnostic Imaging and Radiology, Children’s National Hospital, 111 Michigan Ave NW, Washington, DC 20010, USA; 13Newcastle University Centre for Cancer, Translational and Clinical Research Institute, Newcastle University, Newcastle Upon Tyne NE1 7RU, UK; 14Princess Máxima Center for Pediatric Oncology, Heidelberglaan 25, 3584 CS Utrecht, The Netherlands

**Keywords:** drug delivery, blood–brain barrier, brain tumor model, preclinical, xenograft, companion animal, childhood brain tumors, drug repurposing

## Abstract

**Simple Summary:**

Brain tumors account for over 20% of childhood cancers and are the biggest cancer killer in children and young adults. Several initiatives over the past 40 years have tried to identify more effective drug treatments, but with very limited success. This is largely due to the blood–brain barrier, which restricts the entry of many drugs into the brain. In this review, we describe the main techniques that are being developed to enhance brain tumor drug delivery and explore the preclinical brain tumor models that are essential for translational development of these techniques. We also identify existing approved drugs that, if coupled with an efficient delivery method, could have potential as brain tumor treatments. Bringing this information together is part of a funded initiative to highlight drug delivery as a research strategy to overcome the current challenges for children diagnosed with brain tumors.

**Abstract:**

Brain and spinal tumors affect 1 in 1000 people by 25 years of age, and have diverse histological, biological, anatomical and dissemination characteristics. A mortality of 30–40% means the majority are cured, although two-thirds have life-long disability, linked to accumulated brain injury that is acquired prior to diagnosis, and after surgery or chemo-radiotherapy. Only four drugs have been licensed globally for brain tumors in 40 years and only one for children. Most new cancer drugs in clinical trials do not cross the blood–brain barrier (BBB). Techniques to enhance brain tumor drug delivery are explored in this review, and cover those that augment penetration of the BBB, and those that bypass the BBB. Developing appropriate delivery techniques could improve patient outcomes by ensuring efficacious drug exposure to tumors (including those that are drug-resistant), reducing systemic toxicities and targeting leptomeningeal metastases. Together, this drug delivery strategy seeks to enhance the efficacy of new drugs and enable re-evaluation of existing drugs that might have previously failed because of inadequate delivery. A literature review of repurposed drugs is reported, and a range of preclinical brain tumor models available for translational development are explored.

## 1. Introduction

Annual global incident cases of cancer in children and young people (CYP) aged 0–19 years are estimated at around 400,000 [1,2]. Childhood malignancies are classified into 12 major categories, grouped by the tissues of origin [3]. Tumors of the central nervous system (CNS) are the most common solid tumors in children between 0 and 14 years, accounting for 10–37% of all cancers across 0–19 years of age. The age groupings with the highest and lowest incidence frequencies are the 5–9- and 15–19-year-old age groups, respectively [4].

The world age-standardized incidence rate (WSR) for brain tumors at 0–14 years of age is 28.2 per million person years, being higher in European countries (WSR 30–38.9) and lowest in sub-Saharan Africa (WSR 6.3). The WSR among young people aged 15–19 years is 19.9 per million person years, ranging from 6.0 to 36.2 per million person years [4]. The presentation of these tumors along with their histology and molecular and anatomical characteristics is outlined in Figure 1 and Appendix A.

Based on Global Cancer Observatory estimates of incidence rates for European countries, the cumulative risk of a child developing a CNS tumor is 1 in 6670 by 5 years of age, 1 in 3700 by 10 years of age, 1 in 2670 by 15 years of age and 1 in 2080 by 20 years of age [5]. In upper-middle income and high-income countries, the current survival rates (Figure 1) have been achieved by combining surgery, radiotherapy and drug therapy. The presentation of these tumors along with their histology and molecular and anatomical characteristics is outlined in Figure 1 and Appendix A.

The evolving molecular data of childhood brain tumors have far outpaced success in translating these scientific results into clinical trials. Though genomic analysis has revolutionized the classification of these malignancies, the major challenge lies ahead in identifying treatment strategies and their relationship to the revised classification [6,7]. Any trial of a new therapy would need to consider the role of drug delivery and the capacity to achieve effective and non-toxic concentrations of drugs administrated to tumor tissue and healthy brains.

Treatment for brain tumors can be assessed in terms of three tumor situations. First, we consider primarily resistant tumor types with very poor survival rates. For these tumors, experimental interventions are directed at the primary tumor, and the aim is to develop delivery techniques that ensure adequate drug exposure at the tumor site. Examples include diffuse midline glioma (DMG), ependymoma, atypical teratoid rhabdoid tumor (ATRT), high-grade glioma (HGG) and malignant rare variants. Second, we consider malignant tumor types with established sensitivity to drug therapy. For these tumors, treatment of metastatic disease is currently reliant upon extended-field radiotherapy. A promising alternative is the use of intracerebrospinal fluid (CSF) delivery approaches to target leptomeningeal spread [8]. Third, we consider all brain tumors that have not yet become treatment-resistant nor spread to the leptomeninges. The aim here is to target drug delivery to the precise anatomical tumor locations in the brain, to increase effectiveness and reduce systemic toxicities in the developing child.

For innovative approaches to be adopted, it needs to be proven that the extent of drug delivery to the tumor location is sufficient to achieve its therapeutic effect. Unfortunately, most cancer drugs do not penetrate the blood–brain barrier (BBB) effectively, which may explain the recurrent failure that has been the history of brain tumor drug development. Indeed, the BBB presents a significant obstacle to a wide range of anticancer drugs, from cytotoxic agents and small molecules to immunotherapies and antibody–drug conjugates.

It is the problem of the uncertainty of systemically administered drug penetration of the BBB that is the focus of this review. The BBB is structurally complex, composed of endothelial cells, pericytes and astrocytes, forming a neurovascular structural and functional barrier that is highly effective at maintaining homeostatic levels of intracerebrospinal fluid (CSF) concentrations and preventing the influx of circulating pollutants [9,10]. Several factors restrict drug compounds passing the BBB, including endothelial tight junctions and drug-transporter-mediated efflux. Tight junctions create physical links that impair or block molecules in circulation from entering brain parenchyma directly, based on molecular size, lipophilicity, ionization and polarity. Both direct cellular interactions and paracrine signaling from CNS astrocytes and pericytes lead to the formation of tight junctions [11]. Transcellular drug entry is governed by endocytosis, passive diffusion and the ratio of inward- to outward-facing membrane drug transporters [12]. The BBB surrounding capillaries in the brain parenchyma is then modified to form a blood–tumor barrier upon the initiation and progression of primary brain (or metastatic) tumors [13]. This barrier thus limits the delivery of systemically administered drugs, a physiologic dilemma that has limited the therapeutic efficacy of chemotherapies directed toward pediatric tumors. Failures in chemotherapy trials can be partially attributed to poor drug delivery, as fewer than 5% of chemical compounds have been shown to achieve therapeutic concentrations in the CNS [14,15]. Consequently, a number of intracranial drug delivery approaches are being pursued to address this issue. We have reviewed the evidence for alternative approaches to CNS drug delivery, which are designed to offer mechanisms of drug delivery that can augment penetration of, or bypass, the BBB to ensure that drugs’ tissue levels reach effective concentrations. These include selecting drugs with BBB-permeable properties, intra-arterial chemotherapy, BBB disruption, intrathecal/intraventricular/interstitial administration and nano-carrier delivery vehicles (Figure 2).

Furthermore, existing approved drugs might have been discounted for development as a brain tumor treatment due to their inability to reach the brain. By harnessing the new delivery techniques that are in development, these should now be reconsidered and evaluated as potential brain tumor treatments [16]. The World Health Organization (WHO) has recently made a powerful economic case for commissioning child cancer therapies globally to exploit a 3:1 economic payback in societal terms [17]. Repurposing approved drugs with drug delivery techniques offers an attractive economic strategy for innovation. The CNS drug delivery systems in development would also be compatible with new molecularly targeted drugs emerging from the cancer drug pipeline.

In considering the exciting challenge of the development of these new personalized therapies, there is much to be learnt about their specificity and efficacy of action as well as the risk of toxicity. This is particularly important given that their likely targets are brain development mechanisms in tumors that occur in patients whose brains are still developing. To optimize their potential, a key element of theoretical development will be to ensure that they are deliverable to the brain tissue in controlled concentrations to offer efficacy and control toxicity. To avoid the risk of recurrent failure for unexplained reasons, which has been the history of brain tumor drug development, we propose that consideration is needed specifically for drug delivery in this new era. In this review, the focus will be upon the physical and pharmaceutical methods that exist to deliver drugs across the BBB, loco-regionally. We also describe the range of existing drugs suitable for repurposing using these techniques and how such developments might be prepared for trial using models mimicking the drug delivery steps to the brain and the tumor.

**Figure 2 cancers-15-00857-f002:**
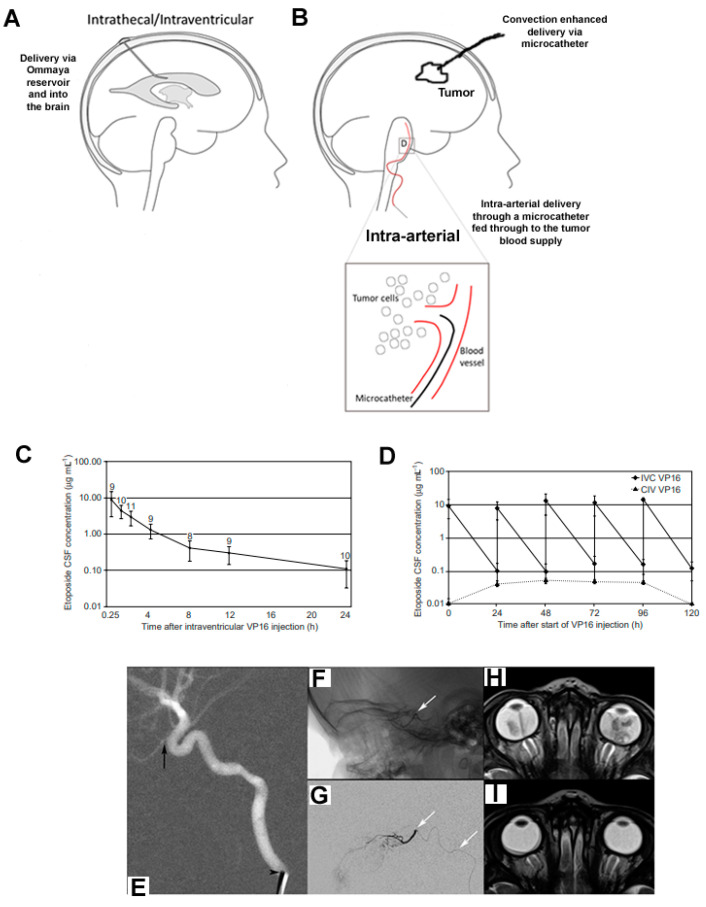
Intracranial drug delivery methods. Several methods can be utilized to maximize drug delivery to brain tumors: (**A**) intrathecal/intraventricular delivery via an Omaya reservoir (catheter) to administer drugs directly to the cerebrospinal fluid by the ventricle or subarachnoid space in the spine; (**B**) interstitial delivery using microcatheters and convection-enhanced delivery; (**C**,**D**) intra-arterial administration; (**C**) CSF concentration time profile following intraventricular etoposide administration (0.5 mg) on the first day of a 5-day schedule (mean ± standard deviation, in a total of 11 courses in 4 patients; the number of measurements is given above each data point); (**D**) CSF concentrations (mean ± standard deviation) following intraventricular (IVC) etoposide administration (0.5 mg per day) on 5 consecutive days in the main study group (◆) (peak and trough levels, 15 courses in 5 patients) compared with CSF concentration following continuous intravenous infusion (CIV, 400 mg m-2 over 96 h) in the second experimental group (▲) (trough and steady-state levels, 5 courses in 2 patients); (**E**–**I**) intra-arterial chemotherapy (IAC) for retinoblastoma-super-selective catheterization and MRI before and after three sessions of IAC: (**E**) lateral roadmap shows a 4-French catheter (arrowhead) in the cervical ICA of the ophthalmic artery (arrow); (**F**) unsubtracted and (**G**) subtracted lateral views from a super-selective ophthalmic artery catheterization (white arrows denote the microcatheter, whose tip is at the ophthalmic artery origin); (**H**) axial T2 MRI of the orbits prior to IAC and (**I**) after 3 cycles show a marked reduction in tumor volume bilaterally. Figure 2E–I reprinted with permission from [18]. Copyright year 2023; copyright owner, Ruman Rahman [18].

## 2. Overview of CNS Drug Delivery Methods

Extensive research has generated clinical trials of existing cytotoxic chemotherapy agents and new, biologically targeted drugs, and is ongoing. Despite this significant effort, there are only four drugs approved for use in brain tumors by the Food and Drug Administration (US) and European Medicines Agency: lomustine (1-(2-chloroethyl)-3-cyclohexyl-1-nitrosourea; CCNU) [19], temozolomide [20], carmustine wafers [21] and everolimus [22]. CCNU and temozolomide are alkylating agents with physicochemical properties permitting enhanced access across the BBB. Carmustine wafers are placed in tumor cavities at resection. Everolimus is an mTOR inhibitor that targets the mutated mTOR pathway in tuberous sclerosis where tissue overgrowth in brain tissue can lead to a type of brain tumor called sub-ependymal giant cell astrocytoma (SEGA) [23]. It is licensed for treatment of SEGA in children and young adults where the tumor cannot be safely resected. Three of these four approved drugs, therefore, were selected for their properties to either cross or bypass the BBB. All other drugs in use for the treatment of brain tumors in children are either experimental or used off-label, including regimens containing vinca alkaloids, alkylating agents, topoisomerase inhibitors, folic acid antagonists, anthracyclines and anti-angiogenic agents [24].

The uncertainty surrounding how systemically administered drugs penetrate the BBB is the problem that is the focus of this review. We have reviewed evidence for alternative approaches to CNS drug delivery that are designed to offer mechanisms of drug delivery that can augment penetration of, or bypass, the BBB to ensure that drug tissue levels reach effective concentrations. Methods for augmenting penetration of the BBB include intra-arterial administration, and pharmacological and ultrasound modulation of the BBB. Methods directed at bypassing the BBB include intra-CSF and interstitial administration using drug infusions (convection-enhanced delivery) or carrier polymer systems.

## 3. Augmenting Drug Passage through the BBB

### 3.1. Pharmacological Modulation of the BBB

The innate pharmacokinetic properties of a particular chemotherapy agent, which facilitate its passage through the BBB and thereby its delivery into brain tissue, are vital for its therapeutic efficacy. Drugs that were more likely to permeate the BBB included compounds that had a molecular weight of ≤500 Da, were lipophilic (logP (partition coefficient) > 1), were relatively nonpolar (≤5 hydrogen bond donors, ≤10 hydrogen bond acceptors) and had <90% protein binding (thereby facilitating unbound drugs to cross the BBB) [25].The presence of luminal membrane transporters on brain capillaries commonly results in efflux of a limited number of lipophilic drugs possessing the appropriate physicochemical properties to achieve meaningful BBB penetration [26].

Rather than tailoring the physiochemical properties of the drug to augment BBB penetration, other approaches attempt to increase endothelial cell permeability using bradykinin and its analogues or drugs that inhibit or modulate P-glycoprotein (P-gp) and other relevant drug transporters in the brain [27]. The use of the bradykinin analogue lobradimil in combination with the widely used anticancer drug carboplatin was investigated by the US Children’s Oncology Group, following promising results generated in a glioma preclinical model [28,29]. While this phase II trial failed to show any significant clinical benefit in pediatric brain tumor patients, it represents a potentially useful approach if further advances can be made in this area.

Research over many years has resulted in development of selective and specific P-gp modulators such as tariquidar, which have the potential to show clinical benefits in combination with both well-established and novel targeted anticancer drugs. In this respect, early-phase clinical trials have shown the feasibility of administering tolerable and biologically active doses of tariquidar in conjunction with doxorubicin, docetaxel and vinorelbine in a pediatric cancer setting [30]. As most newer targeted anticancer drugs are not substrates for P-gp or related transporters, they may offer a more rational approach for drug combination studies.

### 3.2. Ultrasound-Induced BBB Disruption

Ultrasound (US)-induced opening of the BBB was first described by Bakay et al. in 1956 [31]. Hynynen et al. later demonstrated that the intravenous injection of microbubbles prior to pulsed low-intensity US allowed for a reduction in the acoustic pressure necessary to safely open the BBB [32]. By inducing expansion and contraction of injected microbubbles (cavitation), four different cellular mechanisms may occur, leading to an increased transport of drugs across the BBB: transcytosis; transendothelial fenestrations; opening of tight junctions; and free passage of molecules through the permeable endothelium [33].

The size of BBB opening is dependent on both the acoustic pressure and microbubble size. This procedure has been shown to enhance the delivery of a large variety of therapeutic agents, from small-molecular-weight drugs [34] to monoclonal antibodies [35] or cells [36]. Loaded liposomes and microbubbles [37] have been used to optimize local drug delivery. Different murine tumor models have been exposed to systemic drugs after US-induced opening of the BBB, and both tumor control and increased survival have been achieved [38,39].

Monitoring of BBB disruption can be performed using magnetic resonance imaging (MRI), where BBB disruption appears as a leakage of contrast agent (gadolinium) in T1-weighted sequences. The focal signal enhancement is dependent on the applied acoustic pressure. A method based on acoustic emission control has been developed for use with transcranial-focused US devices. With this method, acoustic pressures are adjusted in real time based on monitoring bubble activity. The technique can potentially be used to safely modulate BBB disruption [40].

Transcranial, noninvasive systems have been developed, such as the ExAblate^®^ 4000 system (InSightec, Haifa, Israel) [41], and an alternative, implantable ultrasound device, the SonoCloud^®^, has been recently developed by CarThera SAS (Paris, France) [42]. The safety of both the ExAblate^®^ system (NCT02343991) and the SonoCloud^®^ device (NCT02253212) have been recently confirmed in clinical trials with encouraging results [43,44].

### 3.3. Intra-Arterial Chemotherapy

The intra-arterial (IA) route for drug delivery to brain tumors has been explored since the early 1950s [45,46] (Figure 2C,D). IA chemotherapy for brain tumors is administered through a catheter inserted into the carotid or vertebral artery. The drugs then travel through the capillary networks of the bloodstream and, eventually, into brain tissue. Studies to date have shown that the intra-arterial route is significantly more effective for drug delivery to the brain than the intravenous route (also via the bloodstream). To target treatment more specifically to a brain tumor, chemotherapy can be infused into the main tumor-supplying arteries rather than into the carotid or vertebral arteries. This is called super-selective intra-arterial chemotherapy (IAC).

Super-selective IAC is safe and technically feasible due to advancements in microcatheter design and imaging systems that facilitate navigation through the delicate intracranial vasculature. Selective IA cerebral infusion (SIACI) of chemotherapy is a technique designed to selectively increase the local concentration of a drug in the peri-tumoral vascular supply, thereby bypassing first-pass metabolism. Current studies are directed toward utilizing SIAC in conjunction with methods to disrupt the BBB, as there is little evidence to show that SIACI can be effective alone. IAC for retinoblastoma has been the most widely accepted and successful neuro-oncologic application of IAC and is now a standard treatment option for retinoblastoma (Figure 2E–I).

IAC clinical trials for brain tumors thus far are diverse (Appendix A), and analyzing their results is challenging because of the small numbers of participants enrolled in each study and the various therapies each patient has received prior to enrolment [47]. More recently, IA BBB opening with mannitol followed by high-dose IA bevacizumab has been found to be at least as effective as systemic chemotherapy for the treatment of recurrent glioblastoma, with limited systemic toxicity [48]. Recently published positron emission tomography (PET) studies in mice further demonstrate the advantage of the IA route, which is far more effective for bevacizumab delivery to the brain when compared with the intravenous route [49].

More precise IA drug delivery can be aided by quantitative visualization of BBB opening. Novel hybrid imaging platforms, specifically multimodality imaging suites that combine X-ray with MRI, or X-ray with MRI and PET-CT, are being developed to fully utilize the capabilities of each imaging modality. Using ultrafast advanced MRI techniques, trans-catheter parenchymal contrast agent flow can be visualized prior to mannitol or therapeutic agent administration, enabling modifications to infusion rate and microcatheter position for more precise BBB opening and subsequent IA drug delivery [50].

## 4. Bypassing the BBB

### 4.1. Polymer Therapeutics for Local Delivery

The potential benefits of local treatment are significant as they address major shortcomings associated with systemic delivery [51]. These shortcomings include low drug concentration at tumor site and systemic dose-limiting toxicities, as well as low efficacy due to rapid inactivation while circulating in the bloodstream. The concept of local delivery of anticancer therapeutics immediately post-surgical resection is based on embedding the drug in a biodegradable biomaterial or polymer for controlled release. A multitude of implantable materials have been developed for that purpose, which can be prepared as wafers, discs, films, rods, particles, meshes/scaffolds or injectable hydrogels [52]. Conventional fabrication techniques include electrospinning, solvent casting, spray drying, freeze drying, extrusion and compression molding.

Biodegradable copolymers impregnated with the alkylating agent carmustine (BCNU) (Arbor Pharmaceuticals, Atlanta, GA, USA) are the only approved drug delivery implant for local treatment of high-grade glioma. A phase III multicenter, double-blind trial in recurrent adult isocitrate dehydrogenase wild-type glioblastoma patients demonstrated improved overall survival from 23 to 31 weeks [53]. While this is clear benefit, the effect was modest and likely due to developing resistance towards BCNU and insufficient diffusion into brain parenchyma beyond the infiltrative margin.

Extensive work in this area has continued over the decades with a breadth of biomaterials used in combination with various antitumor agents. Paclimer microparticles, a sustained-release formulation of paclitaxel-loaded (10% *wt*/*wt*) polyphosphoester particles, prepared as wafers, have been investigated as a potential localized treatment for malignant glioma. Sustained release of active drug was observed for 30 days, doubling the median survival of glioma-implanted rats [54]. Another formulation of polymer evaluated for paclitaxel delivery is an injectable system based on a poly (ethylene glycol)-ϵ-polycaprolactone (MPEG-PCL) diblock copolymer gel. This is a thermosensitive, water-soluble biopolymer that is designed to undergo a reversible thermal gelation upon injection at the tumor site [55]. The therapeutic benefit with these local polymer–drug approaches remains limited, with suboptimal diffusion of therapeutics from the polymer site into brain parenchyma representing a considerable challenge.

Optimized formulations of polymers have been developed for improved diffusion throughout the cerebral/brain tumor interstitium. Examples of such approaches are nanocomplexes composed of DNA condensed into a blend of biodegradable polymer, poly(β-amino ester) (PBAE), with PBAE conjugated with 5 kDa polyethylene glycol (PEG) molecules (PBAE-PEG) [56].

Incorporation of BBB-permeating moieties into the structure of polymer therapeutics is highly desirable. Receptor-mediated transcytosis is one of the most frequently applied strategies, exploiting mechanisms for active transcellular transport of hormones or growth factors [57]. Polymers are furnished with ligands of receptors that are highly expressed on the BBB, such as the transferrin receptor or low-density lipoprotein receptors.

Transferrin-conjugated silica nanoparticles have been successfully used to enhance the delivery of doxorubicin and paclitaxel with high activity against glioma in mouse U87 xenograft model [58]. In similar context, monoclonal anti-transferrin receptor antibody 8D3 has been used for shuttling short hairpin RNA-laden pegylated liposomes, resulting in improved survival of mice implanted with U87 glioma [59].

Targeting low-density lipoprotein-receptor-related protein 1 (LRP1) recently gained interest because of its high expression in both the BBB and glioma. Angiopep-2, a molecule that specifically binds to LRP1, has been used for targeting doxorubicine-loaded nanotubes, improving tumor-targeting efficiency and therapeutic efficacy in a mouse C6 glioma [60]. Another approach to developing polymer therapeutics is to use inhibitors of efflux transporters [61]. Interestingly, anti-SSTR2-peptide-based targeted delivery of PLGA-encapsulated 3,3′-diindolylmethane nanoparticles allowed overcoming the BBB and showed to prevent glioma progression [62].

### 4.2. Convection-Enhanced Delivery

In 1994, the concept of convection-enhanced delivery (CED) was introduced as a solution to the BBB obstacle, enabling direct delivery of therapeutic drugs to the CNS [42]. The drug is infused at precisely controlled infusion rates with the aim of creating a pressure gradient at the tip of an implanted intraparenchymal catheter. This positive pressure drives fluid out from the catheter tip through the extracellular space, with the aim of replacing the extracellular fluid with infusate (Figure 3).

CED has several potential advantages over conventional systemic drug delivery methods and other novel methods of bypassing the BBB. CED facilitates highly accurate anatomical targeting and delivery of higher drug concentrations throughout clinically relevant volumes of brain tissue or tumor. Direct administration means that a negligible concentration of drug enters the systemic circulation, meaning that theoretically high local drug concentrations can be achieved without causing any associated systemic toxicity. CED enables the controlled, homogeneous distribution of drugs through large brain volumes, offering an opportunity to manipulate the extracellular environment of intrinsic malignant brain tumors [63,64]. A list of published clinical trials and clinical reports of CED in high-grade glioma is provided in Appendix A.

#### 4.2.1. Translation of the Technique from Bench to Bedside

Lack of clinical translation thus far may be attributable to problems with CED catheter design and inaccurate catheter implantation techniques. In addition, the physical and chemical properties of the infused drug may make it a poor candidate for CED and, along with poor catheter performance, may cause poor drug distribution [43]. Many drugs that have been investigated for use via CED are poorly understood in terms of their pharmacokinetics and pharmacodynamics. Accurate preclinical analysis of drug distribution in large animal models of CED is of paramount importance prior to clinical translation.

#### 4.2.2. Catheter Design

One major obstacle to distributing drugs effectively is the reflux of infusate along the catheter tract at the catheter–brain interface [63,65] There are three main elements that are thought to influence the rate of catheter-related reflux; the catheter diameter, tissue trauma on catheter implantation and the speed of catheter insertion. Catheter design has since developed to incorporate a “step,” where there is a drop from a large to a small diameter proximal to the catheter tip. The mechanisms by which a step design reduces reflux may be due to focal compression of tissue at the step–catheter interface, which effectively creates a seal. The step design has been developed further with the design of a novel “recessed-step” catheter, which has shown superior reflux resistance to a conventional stepped catheter, both in vitro and in vivo [66].

#### 4.2.3. Convection-Enhanced Delivery Infusion Regimes

The ability to control and titrate the rate of infusion is fundamentally important for successful CED. Stepped catheters have enabled reflux-free infusions to be performed in translational and clinical studies of CED at flow rates of up to 10 μL/min [66]. Higher flow rates enable the duration of infusions to be reduced to timescales that are acceptable to patients. There is, however, a balance between using high infusion rates and increasing the risk of trauma because of excess pressure at the catheter tip. The maximum safe infusion rate for CED is dependent upon the diameter of the catheter used and the physical tissue properties of the target structure in the brain.

#### 4.2.4. Intermittent Convection-Enhanced Delivery

The ability to repeatedly administer drugs via CED to the same target volume without the need for further surgery is especially important when treating malignant tumors. All glioma CED clinical trials published to date have used a temporary implanted catheter, and further cycles of treatment have required repeat implant surgery [67]. Chronic drug delivery is also complicated by the potential for drug–tissue binding at the catheter tip, local toxicity and low infusion rates, which all result in suboptimal drug distribution [68]. The development of a novel implantable CED drug delivery system comprising up to four intracranial-implanted recessed-step catheters connected to a transcutaneous, septum-sealed bone-anchored port has made intermittent CED possible, without the need for further surgery [69].

#### 4.2.5. Drug Properties Required for Convection-Enhanced Delivery

CED exploits bulk flow through the interstitial spaces of the brain, which allows drugs with a wide range of molecular weights, including nanoparticles and viral constructs, to be effectively distributed within the brain parenchyma. The surface properties and tissue affinity of an infused molecule are far more important than molecular weight in influencing its distribution in the brain parenchyma after administration via CED. The ideal drug suitable for CED, therefore, should be water-soluble, of neutral or anionic charge, and have low tissue affinity or binding, but these same characteristics may result in rapid interstitial clearance from the brain through perivascular spaces into the venous sinuses or CSF [70]. Water insoluble drugs, such as the histone deacetylase inhibitor panobinostat, may be delivered via CED when “packaged” within a nanoparticle, such as a polymeric micelle. This novel strategy of combining CED with nanoparticle drug delivery has been effective preclinically in glioma models and has potential for clinical translation [71].

A confounding factor for CED is the short brain half-life of many chemotherapeutics and the subsequent post-infusion rapid clearance. Drug encapsulation within nanoparticles offers a means for both enhanced stability and controlled release, prolonging the half-life of the chemotherapeutic, and hence increasing tumor cytotoxicity [72,73]. Drug encapsulation in principle should reduce neurotoxicity and increase tissue retention. Real-time PET has been used to demonstrate prolonged nanofiber-bound drug retention in situ relative to free drug [74]. Polylactic-co-glycolic acid nanoparticles encapsulating carboplatin [75], paclitaxel [76] and camptothecin [77] have shown improved sustained drug release relative to free drug alone. Further studies have explored nanoparticles bearing dual functionality for CED delivery, whereby nanoparticles containing a magnetic core can additionally carry drug cargo, offering theranostic capability in vivo [78,79].

### 4.3. Intra-CSF or Interstitial Administration

Leptomeningeal malignancy complicates up to 55% of childhood cancers, including brain tumors, and represents a rate-limiting step to cure [8,80]. In brain tumors, dissemination from the primary tumor, before or after surgery, via CSF pathways is assumed. However, evidence exists to support the vascular route of dissemination. For primary brain tumors, the standard therapy is craniospinal radiotherapy, but the attendant risk of acute and delayed brain injury and endocrine deficiencies compounds post-radiation impairment of spinal growth. Alternative ways of treating leptomeninges by intensifying drug therapy delivered to CSF are being investigated [8,81]. Current methods of bolus administration are complex and burdensome clinically. There is a need to establish devices and techniques to deliver intra-CSF therapy more easily, especially if prolonged infusions or sustained release preparations are to be developed. Sharing the development of such delivery systems and testing repurposed drugs with the needs of adult practice would create synergy for their commercial development.

## 5. Repurposing Drugs in Pediatric Neuro-Oncology

It is thought that many drugs have been discounted as potential agents for treating brain tumors because they are known to be unable to cross the BBB. Additionally, it is highly likely that experimental drugs that have been labeled as ineffective were considered so as a consequence of inadequate drug concentration at the tumor site. It is therefore imperative to evaluate potential therapeutic agents via an approach that ensures adequate delivery to the tumor.

Repurposing, or repositioning, refers to the alternative use of existing approved drugs for different diseases. Medications go through strict approval procedures, carried out by the Food and Administration (FDA) and/or European Medicines Agency (EMA), to ensure drugs are safe and effective, and approval is usually granted for the use of a drug for a specific condition. The drug discovery process for novel therapeutics requires a vast financial input. The full capitalized cost per new approved compound was averaged at USD 1.2 billion, with the disadvantages of slow market approvals and only a 12% probability of clinical success [82]. By utilizing approved drugs with known pharmacokinetics and detailed toxicities and tolerance, repurposing offers a fast-tracked, cost-effective approach to develop new treatments.

A PubMed search using the term “drug screen pediatric brain tumors” identified twenty-one articles published in the past five years, of which seven met the criteria for inclusion in this review: full-text availability; original research article; drugs classed as repurposed; data validated in pediatric brain tumor models. Together, these publications assessed a total number of 282 compounds against pediatric brain tumors, including HGG/DMG, medulloblastoma and embryonal tumors with multilayered rosettes (ETMRs) (Figure 4). Drug databases (Appendix A) were used to determine the approval status and mode of action of each drug. Forty-three percent of drugs tested have been approved for use in humans (Appendix A), and an additional one percent were approved as brain tumor therapeutics.

### Anti-Helminthic/Psychotic/Seizure Drugs

Microtubule inhibitors have been used for the treatment of both pediatric and adult brain tumors, despite a lack of evidence of their efficacy as a monotherapy in either animal models or clinically [89,90]. The primary reason for this poor efficacy appears to be a lack of BBB permeability [91,92]. Moreover, microtubule drugs, such as vincristine, tend to have severe, dose-limiting toxicities due to cumulative neurotoxicity [93].

Mebendazole was serendipitously found to be active against high-grade astrocytoma by the Riggins group at Johns Hopkins University, when their mouse colony was infected by pinworms and treated with fenbendazole, an analog of mebendazole, causing a strong inhibition in tumor take rate [94]. Subsequently, mebendazole was shown to be highly active in several orthotopic models of medulloblastoma [95].

The probable mechanism of action of mebendazole is through inhibition of microtubule polymerization [89,94]. Generally, mebendazole is well tolerated with few side effects [96]. Two clinical trials examining the safety of mebendazole for the treatment of pediatric brain tumors are ongoing: a phase I/II study examining mebendazole in combination with vincristine, carboplatin and temozolomide in pediatric high-grade glioma patients (NCT01837862); and a phase I study examining the safety of mebendazole monotherapy in a wider range of pediatric brain tumor patients, including medulloblastoma (NCT02644291).

The voltage-gated potassium channel, EAG2, is enriched on the trailing edge of migrating medulloblastoma cells and facilitates cell motility [97]. Subsequently, researchers tested a number of approved drugs in the in vitro growth of medulloblastoma cell lines, which led to the discovery of the novel EAG2-blocking action of the anti-psychotic drug thioridazine [97]. This action was accompanied by a reduction in medulloblastoma tumor growth and metastasis [97].

Epigenetic modifying drugs such as the histone deacetylase (HDAC) inhibitors panobinostat and sodium valproate have shown high efficacy against DMG and ETMR. In medulloblastoma, sodium valproate reduced the clonogenicity of cells in vitro, and exerted additive effects in combination with 5-aza-2′-deoxycytidine [98].

Clinical translation has led to the utilization of sodium valproate as an adjuvant chemotherapeutic in several clinical studies, notably in six trials studying its effects in brain tumor patients, three of which were for the treatment of childhood brain tumors (NCT00879437, NCT00107458, NCT03243461). Importantly, sodium valproate has been reported to be BBB-permeable [99], yet only 15% of serum levels reach the brain. A retrospective study demonstrated that pediatric HGG patients who were taking valproate as an anti-epileptic exhibited no additive toxicity when the drug was combined with radio-chemotherapy, and determined that valproate is well tolerated in pediatric HGG patients [100]. Sodium valproate is currently being delivered via CED in early trials, where it is being directly administered to the brain tumor via microcatheters [101].

## 6. Brain Tumor Models

### 6.1. Human-Specific In Vitro Brain Tumor Models

In vitro models are driving forces of drug discovery and remain a mainstay for rapid screening of putative therapeutic agents for priority selection for in vivo preclinical models (Figure 5). Classical 2D monolayer screens do not necessarily translate accurately to in vivo models and so are limited in their use to identify ideal therapeutic agent candidates for brain tumor drug delivery. By contrast, dynamic 3D cultures offer a more physiological test-bed and may permit crude readout of drug penetration. Three-dimensional cellular structures recapitulate cell–cell and cell–extracellular matrix (ECM) signaling, creating more physiologically relevant niches compared with two-dimensional culture. Hyaluronic-acid-based tissue-engineered matrices have been shown to promote GBM invasion in vitro and thus offer a means to assess anti-invasive therapeutic agents [102,103]. A combination of advanced manufacturing and 3D in vitro tissue models is a new possibility that could allow us to bioprint in vitro models of cancer. In the pediatric setting, co-cultures of medulloblastoma cells with human fetal brain tissue have been used as a platform for therapeutic agent selection; such co-cultures validated etoposide as a candidate for localized drug delivery [104]. In addition, similar 3D pediatric medulloblastoma spheroids have been successfully used to assess penetration of poly(glycerol-adipate) nanoparticles [105]. There is also an increasing appreciation of the value of in silico modeling to facilitate brain tumor drug delivery and expedite TA selection for in vitro and in vivo screens. For example, computational approaches have enabled factors that prevent/facilitate drug diffusion to the brain via systemic or localized delivery to be accurately modeled for a paclitaxel-loaded hydrogel, via mass transport simulations [106]. Other computational simulations have modeled how TA, injection and physiological properties can affect the regional deposition of intra-arterial delivery to the brain [107].

### 6.2. Rodent Models

Since the distance that the drug needs to travel is important in terms of drug delivery, large animal models are more relevant to clinical situations. However, the development of brain cancer models in large animals is not trivial and is costly. Therefore, in many instances, small animal models are still predominantly used for testing the efficacy of therapeutic agents. Whilst different cell types have been used for tumor induction, with much of the research performed using well-established and characterized cancer cell lines, there is progress in using patient-derived xenografts (PDXs), as well as mutagens and genetically induced tumors (Figure 6, top).

From the perspective of drug delivery, rodents are most suitable for testing systemic routes with passive BBB crossing. If a therapeutic agent is actively transported through the BBB, the molecular differences and species-dependent diversity of composition of molecules involved in this process may affect the prognostic value. Rodent vasculature is also potentially amenable to testing intra-arterial delivery, but the lack of access to super-selective catheter placement prevents spatially specific targeting of therapeutic agents to the tumor-bearing brain territory. While convection-enhanced drug delivery could potentially be tested in rodents, the brain size differences may prohibit reasonable clinical translation of results. Whilst there are described accesses to the intrathecal space in rats and mice, they are used only incidentally due to the technical issues related to their narrowness and very limited CSF volumes. At present, rodent xenograft and allograft orthotopic brain tumor models, which are amenable for surgical resection, are the most widely utilized in the evaluation of intra-cavity and direct interstitial delivery as the search for next-generation carmustine-loaded polymer-like technology continues.

#### 6.2.1. Carcinogen-Induced Models of Glioma

The carcinogenic induction of gliomas stemmed from research on tumorigenesis, where the growth of tumors in mouse brain was achieved nearly a century ago, through intracerebral implantation of pieces of sarcomas and carcinomas [108]. Carcinogenic N-nitroso derivatives are characterized by the specific organotropism after systemic delivery, and N-nitrosoureas administered to rats weekly through several months at a low dose almost selectively produced tumors of the nervous system [109]. It is believed that the tumors created by the systemic delivery of carcinogen are more of neuroectodermal origin than those created by local placement of carcinogen, in which all surrounding elements, including those of mesodermal origin, are included. The systemic delivery of carcinogens can also directly mimic the clinical situation of patients exposed to brain-tumor-driving substances. It was shown that a single dose of ethylnitrosourea (ENU) was sufficient to produce brain tumors in offspring. Interestingly, in addition to the visible tumors, there were also multiple microtumors detected histologically. The N-nitrosourea-induced tumors were also shown to be propagated in vitro and successfully re-introduced to the brain [110]. One of these models, namely rat gliosarcoma 9L, has been instrumental in showing the efficacy of systemic administration of such drugs as BCNU [111] and CCNU [112]. While most cell lines derived from carcinogen-induced brain tumors after transplantation grow a well-demarcated mass, the CNS-1 tumor line obtained from rat glioma induced by chronic exposure to ENU extensively infiltrates the brains of recipients [113]. Importantly, carcinogen-induced tumors are still routinely used in preclinical research (Figure 6, top).

#### 6.2.2. Genetically Engineered Models of Brain Cancer

Pioneering work performed over half a century ago revealed that simple subcutaneous inoculation of mastomys neonates with the SV40 virus produced papillary ependymomas in a majority of animals 3–8 months later without any abnormality at the site of implantation [114]. Then, it was shown that intracerebral injection of Rous sarcoma virus as well SV40 virus produces glial cell tumors in hamsters [115]. It has recently been shown that retrovirally induced overexpression of platelet-derived growth factor-β (PDGF-β) in brain stem glial progenitors is sufficient to produce tumors, which are slow-growing and can correspond to human pontine gliomas (Figure 6, top) [116].

There is, in particular, a lot of interest in next-generation transgenic mice utilizing the RCAS/t-va system. The t-va receptor for subgroup A avian leucosis virus (ASLV), under a desired promoter such as nestin or glial fibrillary acidic protein (GFAP), can be cell-specifically transduced using replication-competent ALV splice acceptor (RCAS) viral vectors derived from ASLVs, which are genetically modified to accept insertion of various oncogenes of interest [117]. The use of the Barh1 homeobox gene promoter to express tv-a was sufficient to produce medulloblastomas after intracerebellar delivery of the active N-terminal fragment of sonic hedgehog (SHH) and a stabilized N-myc proto-oncogene protein (MYCN) mutant in an RCAS vector [118]. A key advantage of using conditional systems is an inherent flexibility regarding the choice of oncogenes to produce tumors, as different genetic alterations are typically present in pediatric and adult brain tumors.

#### 6.2.3. Patient-Derived Xenografts

The implantation of tumor fragments directly derived from particular patients into rodent host brains offers a new paradigm of precision medicine (Figure 6, top), wherein patient-derived xenografts (PDX) have permitted combined therapeutic modality studies. However, the clearly defined borders in histological assessment are not aligned with the true infiltrative nature of malignant brain tumors. More recently, the introduction of next-generation sequencing combined with computational modeling has facilitated the genome-wide search for functional targets. PDX models, for example, have been instrumental to confirm in silico and in vitro data and have identified a multi-histone deacetylase inhibitor panobinostat and the histone demethylase inhibitor GSK-J4 as a promising therapeutic strategy for (DMG) [88]. PDXs have also facilitated identification of specific drugs for pediatric GBM with the BRAF mutation V600E being an example where a PDX confirmed the efficacy of BRAF V600E inhibitors [119]. However, because PDXs do not have a functional immune system [117], they cannot be used to evaluate the efficacy of immunotherapies.

### 6.3. Large Animal Models

#### 6.3.1. Swine

The modeling of brain cancer in pigs is highly relevant for studying drug delivery, as exemplified by the growing popularity of pig models for convection-enhanced delivery studies. Neurosurgical resection of brain tumors mimicking the clinical-like scenario is also feasible using mini-pigs, so this model is also amenable for studying local drug delivery devices such as carmustine-loaded wafers that are placed in post-removal cavities. When compared with rodents, there are also higher interspecies physiological similarities between humans and large animals with gyrencephalic brains, such as BBB composition, molecular interactions and immune systems [120]. Therefore, collectively, swine models are very attractive for studying systemic drug delivery routes (Figure 6, bottom).

#### 6.3.2. Rabbit

Swine models are, however, not fit to study the intra-arterial route of drug delivery, as they possess a rete of tiny vessels separating extracranial and intracranial vessels, therefore prohibiting introduction of a catheter to the cerebral arteries. The rabbit is the smallest animal in which clinical-grade intra-arterial devices are capable of entering cerebral arteries and was previously used to study intra-arterial injections to treat brain tumors [121]. The recent addition of MRI guidance to intra-arterial procedures allows for more precise and predictable BBB opening and thus for high-dose delivery of therapeutic agents to be monitored [50]. Importantly, the growth of human glioma has recently been shown in immunosuppressed rabbits [122], which can serve as an excellent model for testing intra-arterial therapies (Figure 6, bottom).

#### 6.3.3. Companion Animals

Dogs and cats are the most frequent companion animals, and they can contribute to the development and human translation of novel approaches for the treatment of brain tumors. Their potential has been substantiated by the creation of a comparative brain tumor consortium to focus on the translation of new knowledge from canine to human brain tumor patients [123]. There are 125.8 million households in the USA, with 0.57 dog per household, which gives a total number of 72.5 million companion dogs in the USA alone. With an incidence of approximately 14.5/100,000 for intracranial tumors (i.e., over 10,000 dogs) [124], this gives a relatively high number from the perspective of an experimental population. Spontaneously occurring canine glial tumors have already been extensively exploited in preclinical research, for applications such as testing targeted minicells loaded with doxorubicin [125], addition of procaspase-3 activator to temozolomide [126] and the NanoKnife^®^ system for irreversible electroporation [127].

Similarly, the frequency of neoplastic brain lesions is becoming significant in cats older than 5 years [128], and very rare cases of brainstem gliomas have been found in relatively younger cats (3 and 4 years old) [129], which corresponds to the younger age of gliomas at this location in dogs and humans. While the cat brain is three times smaller than that of the dog, it is still three times bigger than the rabbit brain and therefore also amenable for testing drug distribution and efficacy using any route of drug delivery in principle (Figure 6, bottom).

There are other advantages of using companion animals to test therapeutic strategies—no cost of animal acquisition and maintenance after treatment, the large brain, which allows testing of more clinical-like therapeutic scenarios, the similarity of brain anatomy and physiology as well as immune system to humans and the heterogeneity of tumors, which resembles the clinical situation. Importantly, dogs and cats do not have a rete, and are therefore amenable to any drug delivery route, allowing for relatively easy comparison between various routes. The dog has been shown to be a model animal for testing convection-enhanced delivery [130] and a feasible model for delivering stem cells to the brain and spinal cord using an intra-arterial route [131]. While tumor heterogeneity is a biological advantage, it is a statistical limitation and requires a larger number of animals to be treated to sufficiently power the study. However, the short diagnosis-to-death period of 20 days in untreated cats [132] and 30 days in untreated dogs [124] reduces the experimental time required to assess any experimental therapeutic intervention. Overall, the use of companion animals is a very attractive option for advanced preclinical testing as it allows the search for effective therapeutic strategies that could be used in veterinary medicine.

#### 6.3.4. Non-Human Primates

The monkey model of brain cancer was developed over 30 years ago through intracerebral injection of the oncogenic retrovirus, RSV [133], human polyomavirus obtained from the brain of a patient with progressive multifocal leukoencephalopathy and JC virus [134,135]. Currently, naïve non-human primates are typically used for testing the safety of various anti-tumoral therapies (Figure 6, bottom), particularly using primate-specific therapeutic agents such as oncolytic herpes simplex viruses, as well as drug penetration to the CNS. In some circumstances, such as testing primate-specific therapeutic viruses, non-human primates can be a useful platform to assess safety of proposed therapeutic approaches.

## 7. Discussion

We propose that delivering new and repurposed drugs with enhanced anatomical precision offers a new complementary strategy for the bio-targeted drugs that are emerging. We suggest that there are a series of steps that can be followed to deliver such new strategies for children with brain tumors.

### 7.1. Choose the Tumor Type and Clinical Setting for Targeted Treatments

From the tumor types identified in Figure 1 and Appendix A, drug delivery strategies can be conceived to:Overcome primarily drug-resistant types by ensuring predictable tumor tissue levels of existing or repurposed drugs.Target leptomeningeal metastasis with intra-CSF delivery using existing or repurposed drugs to defer or avoid extended-field radiotherapy.Enhance the efficacy and reduce systemic toxicity of molecularly targeted treatments that are shown to be effective in early trials.

This first step is of key importance as it is the motivation for all subsequent steps based upon the perceived clinical benefit to save lives, avoid life-altering and disabling side effects of extended-field radiotherapy and reduce the risk of toxicities of novel drug therapies.

### 7.2. Identify the Preclinical Research and Development Needed for Trial Design

To translate a new drug and delivery system to the clinic requires significant experimental evidence to justify the treatment concept’s design. Progress can be slow, with carmustine wafers taking more than a decade to be taken to market. Adapting existing devices and repurposing existing drugs offer significant savings. It is important to consider the anatomical location of the tumor, along with the pharmacokinetics of the drug and the parameters of the delivery system, to ensure that the system of choice is given the maximal chance of success.

At a preclinical stage, drug delivery systems should seek to utilize technologies that expedite clinical translation and adoption (e.g., either FDA-approved materials or those that satisfy good manufacturing practice (GMP) guidelines). In addition, drug formulations for drug delivery systems should be amenable for scale up and sterilization. To facilitate accurate efficacy studies, appropriate preclinical models should be matched to the drug delivery system being assessed (e.g., immuno-compromised or immuno-competent).

To achieve realistic transition from the preclinical to clinical trial stage, a Team Science approach is necessary from the outset, whereby cancer biologists and material science and biomedical engineers are strategically supported by clinical experts representing the full clinical pathway (neurosurgery, neuro-oncology, neuropathology, neuroradiology), ensuring clinical relevance and accuracy. Support and advice from pharmaceutical, legal, financial and regulatory disciplines can also be crucial. We advocate utilizing first-in-human phase 0 trials to first demonstrate that drugs to be potentially considered for phase 2/3 randomized controlled trials at a later stage can reach the primary tumor mass (or post-surgical residual disease) at therapeutic concentrations and/or induce their intended biological effect.

Of the direct delivery techniques immediately available for development, intra-CSF therapy is at the top of the list as it is established as effective in pediatric neuro-oncology practice (see Appendix A) [1]. Similarly, intra-arterial and CED systems can deliver drugs at predictable concentrations to specific anatomical sites with minimal systemic toxicities. Ultrasound-based BBB disruption offers significant benefits to enhance intra-vascular delivery across the BBB. The development of polymer–drug delivery systems offers easier administration but challenging pathways of development, as each product requires individual regulatory approval.

## 8. Conclusions

The trial development strategy for a drug delivery system will reliably indicate the delivery system’s and the drug’s feasibility, safety and efficacy. Piloting approaches in human trials requires vision, strong clinical skills, supportive translational research environments, funding and expertise. The societal and economic benefit of treating childhood cancer is justified. However, the rarity of specific tumor types and anatomical targets present significant hurdles for translational research initiatives. Working collaboratively is therefore essential, with those involved potentially having to be prepared for close to a lifetime of dedicated commitment, if a successful outcome is to be achieved.

## Figures and Tables

**Figure 1 cancers-15-00857-f001:**
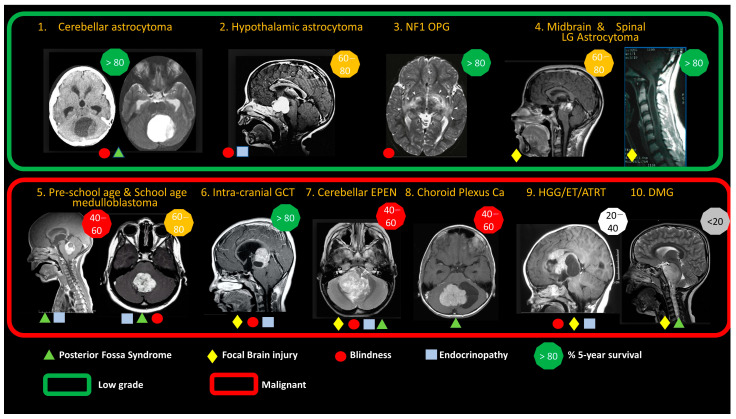
Anatomical features of ten common brain tumors in children, highlighting the predominance of low-grade glioma (green box) and range of malignant tumor types (red box). Low-grade tumors account for ~40% of all childhood brain tumors, are slow-growing and have a very low risk of metastasis. Malignant tumors are fast-growing and have a higher risk of metastasis. Symbols illustrate typical late consequences of the tumor and its treatment, for survivors. Posterior fossa syndrome refers to motor, cognitive and speech consequences of cerebellar mutism syndrome, brainstem damage and prolonged hydrocephalus. Focal injury refers to the risk of other regional focal brain injuries related to tumor growth/invasion of brain structures or the consequences of surgery. Blindness is a consequence of tumor damage to optic nerves, chiasm and tracts or prolonged raised intracranial pressure. Endocrinopathy is due to hypothalamic/pituitary damage from tumor, surgery or radiation therapy to these regions of the brain. The figures illustrate typical population-based 5-year survival rates. See Appendix A for a more detailed description of molecular factors and prognostic criteria. Abbreviations: NF1 OPG, neurofibromatosis type I optic pathway glioma; LG, low-grade; MB, medulloblastoma; GCT, germ cell tumor (germinomatous/non-germinomatous); EPEN, ependymoma; Ca, carcinoma; HGG, high-grade glioma; ET, embryonal tumor; ATRT, atypical teratoid rhabdoid tumor; DMG: diffuse midline glioma. Figure reprinted with permission from textbook “*Brain and Spinal Tumors of Childhood*” under a PLSclear FPL License; Informa UK Limited License No. 78206.

**Figure 3 cancers-15-00857-f003:**
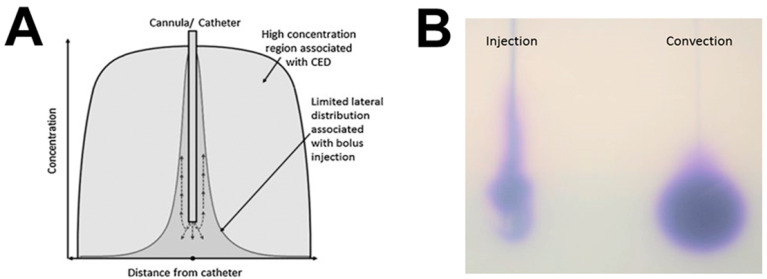
Diagrammatic illustration of the differences in drug distribution achieved by convection-enhanced delivery and intra-parenchymal injection. (**A**) Convection-enhanced delivery is a method of direct infusion of drug into the brain parenchyma under positive pressure with the aim of driving fluid through the extracellular space to cover large regions of parenchyma with a therapeutic drug concentration. In comparison, local injection, which does not create a positive pressure wave, causes local high concentration of the drug to regions of the brain only short distances from the infusion site. (**B**) Pictorial illustration of the difference between injection and convection using a blue dye infused into an agarose gel phantom. Black lines represent the position of the catheter. Injection causes local trauma and poor heterogeneous infusate distribution. Convection-enhanced delivery of the same volume of infusion causes homogenous distribution over a large volume of distribution (unpublished; doctoral thesis of co-author Will Singleton).

**Figure 4 cancers-15-00857-f004:**
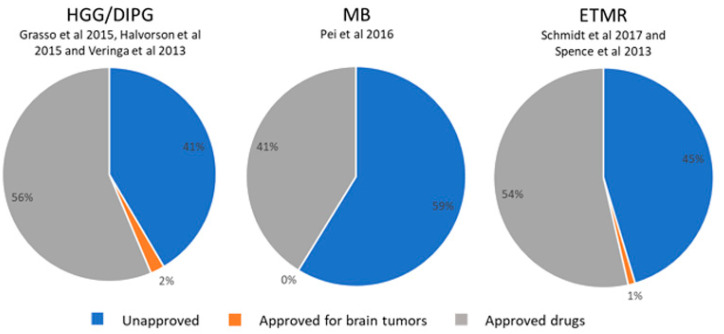
Drug screening data relevant to pediatric brain tumors. A database search identified six publications reporting drug screens conducted on pediatric brain tumor models. These screens were relevant to high-grade glioma (HGG)/diffuse midline glioma (DMG), medulloblastoma (MB) and embryonal tumors with multilayered rosettes (ETMRs). For all studies, most compounds used in the screens were either already approved (41–56% of drugs) or unapproved (41–59% of drugs), suggesting that almost half of drugs tested could be repurposed (original figure) [83,84,85,86,87,88].

**Figure 5 cancers-15-00857-f005:**
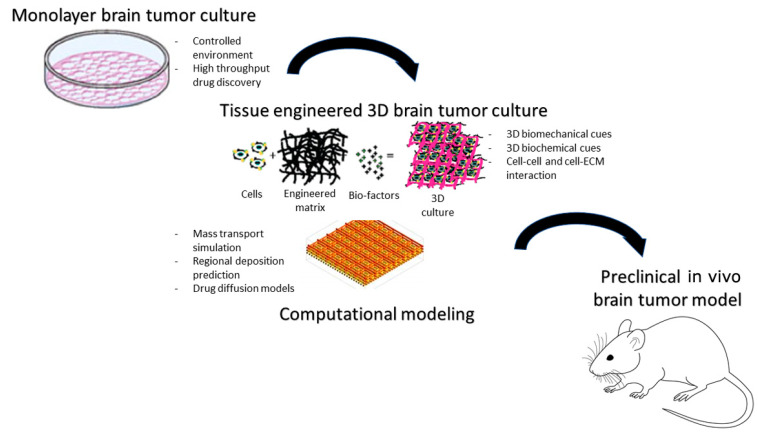
In vitro/in silico predictive brain tumor models identify candidate therapeutic compounds for preclinical drug delivery assessment (original figure).

**Figure 6 cancers-15-00857-f006:**
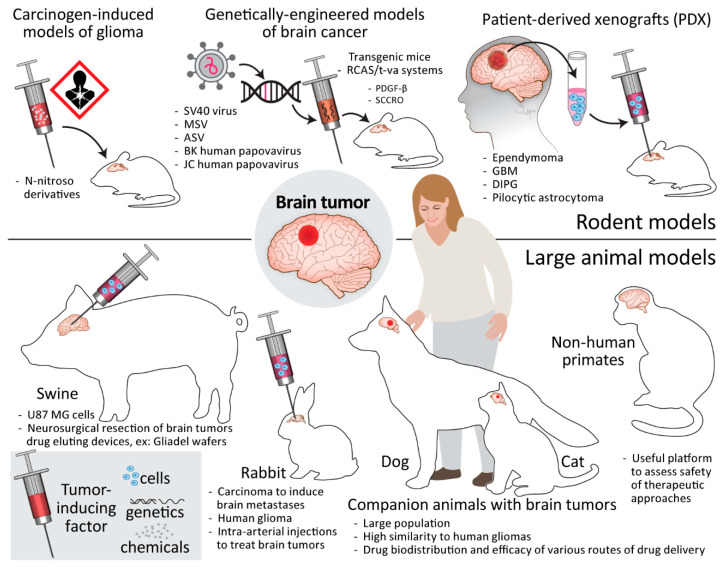
Suite of existing in vivo orthotopic brain tumor models. (Top) Carcinogen-induced, genetically engineered and patient-derived rodent brain tumor models. (Bottom) Large animal models including engineered (swine, rabbit) and naturally occurring de novo animals (dog, non-human primates) (original figure).

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
