# Peer review of "Childhood Brain Tumors: A Review of Strategies to Translate CNS Drug Delivery to Clinical Trials"

_cancers, 2023, doi:10.3390/cancers15030857_

Round 1

Reviewer 1 Report

The review article entitled “Childhood brain tumors: a review of strategies to translate CNS drug delivery to clinical trials” provides a brief description of the currently available techniques enhancing the effectiveness of the clinical treatment of the most common and unfaithful brain tumours occurring in childhood. Considering the low efficacy of the pharmacological protocols it is still necessary to improve the knowledge on this area. The focus of this review is the drug repurposing on the framework of new drug delivery systems to overcome blood-brain barrier limitations. This may be an elegant answer to the clinical needs and deserves to be paid attention to. The overall quality of the text is good, but I suggest reviewing the style that appears non-homogeneous. Some comments are reported below:

·      Considering the review article type, giving the reader a chance to have the broadest perspective is helpful. In light of this, I suggest giving more space to the introduction, and to the critical analysis, of the most relevant works on the topic. Moreover, quite often I noticed that the text lacks a supporting literature reference, as for example in paragraphs: 4.1, 4.2, 4.3. Please, reconsider it.

·       Please make sure that abbreviations have been correctly deciphered, especially when reported for the first time (Lines: 130, 198, 475, 488).

·       I suggest moving figures and tables from supporting information to the main text, it may be in better accordance with the style of a review article.

·       Line 122-134 – This paragraph seems to be not properly linked to the previous argument; in my opinion it might be clearer with a brief introduction.

·       Line 140-151 – As the blood-brain barrier (BBB) is a leading topic of this article, I suggest dedicating a separate and significantly expanded paragraph to it. The anatomical and physio-pathological description of this system may help to understand how to overcome limitations to drug delivery.

·       Figure 2 – In fig. 2A the Ommaya reservoir is cited, but neither in the image caption nor in the main text is its function or definition explained. In fig. 2E-F two graphs from experimental pharmacokinetic observations are reported, similarly in fig. 2G-K some MRI images are shown; please cite the source. Finally, in order to be more attractive, I suggest merging fig. 2B, C and D into one.

·       In line 199 the drugs approved by FDA or EMA for brain tumours are cited: lomustine, temozolomide and everolimus are reported for their active principles name, but DepoCyte (cytarabine) and Gliadel (carmustine) are reported in their commercial name. In order to avoid conflicts of interest, I consider more appropriate indicating all the drugs only for their active principles name.

·       Please make sure that contents reported from line 207 to line 209 were correctly taken from reference 19. This is a clinical commissioning policy of the English National Health Service (NHS) for the use of everolimus in SEGA associated with tuberous sclerosis. It is not reported the off-label use of vinca alkaloids, alkylating agents and the other cited drugs for the treatment of brain tumours in children.

·       Line 222 – I retain that “increase” is not the correct verb for this sentence.

·       I suggest rephrasing the period from line 224 to 230.

·       Line 321 – Please add the object complement to “surgical resection”.

·       Line 323 – I suggest removing the definition “synthetic versus natural” of the implantable materials, in my opinion is not fit.

·       Line 438-439 – Nanoparticles and viral constructs are not possible to define as “complex molecules”.

·       Line 447-449 – I suggest expanding the concept of convention enhanced drug delivery (CED) combined with nanoparticle drug delivery.

·       In figure 5 predictive in vitro / in silico models of brain tumours are shown, but the discussion about in silico models is missing in the main text.

·       Par. 6.2.1. – It might be of interest to deeply investigate about the carcinogen-induced models of glioma (par. 6.2.1).

·       Par. 6.3.3 – I consider that the incidence of brain tumours in cat and dogs may be off topic.

Reviewer 2 Report

In this review, the focus has been upon the physical and pharmaceutical methods that exist to deliver drugs across the BBB, loco-regionally.

In this regard, the authors should deepen the state of the art about the use of other biomaterials and nanosystems for Brain Tumor Drug-Treatment.

Reviewer 3 Report

This is a well written paper on the current concepts in delivery of therapeutics to the CNS. It offers a good overview of current concepts, and presents the authors concrete vision for the coming years.

In section 6 I am missing information on what is known about the blood brain barrier in the different tumor models described. Which models are best for studying drug delivery to the brain, or disruption of blood brain barrier tactics? Is the blood brain barrier in tumors comparable to the normal blood brain barrier? Is this different between the mentioned models? 

In the discussion, I feel advocating the use of Phase 0 trials, to show that drugs used in later Phase 2/3 trials reach the tumor or induce their intended biological effect in brain tumors might be required before starting large trials later phase trials will add to the paper.

Minor comments:

Line 197: Four drugs approved, but the authors continue to sum up five products.

Line 428-430: Needs reference

Line 434: Needs reference to the system mentioned

Line 601: Implantation is misspelled

Round 2

Reviewer 1 Report

The authors significantly improved the quality of the text and kindly addressed each comments. I would like to thank the authors and to suggest the editorial board of Cancers accepting the article in the present, and reviewed, form.